# Gap junction networks in mushroom bodies participate in visual learning and memory in *Drosophila*

**Qingqing Liu[1,2†], Xing Yang[3†], Jingsong Tian[1,2], Zhongbao Gao[1,2], Meng Wang[1], Yan Li[1*], Aike Guo[1,3*]**

[1]State Key Laboratory of Brain and Cognitive Science, Institute of Biophysics, Chinese Academy of Sciences, Beijing, China; [2]University of Chinese Academy of Sciences, Beijing, China; [3]Institute of Neuroscience, State Key Laboratory of Neuroscience, CAS Center for Excellence in Brain Science and Intelligence Technology, Shanghai Institutes for Biological Sciences, CAS, Shanghai, China

**Abstract** Gap junctions are widely distributed in the brains across species and play essential roles in neural information processing. However, the role of gap junctions in insect cognition remains poorly understood. Using a flight simulator paradigm and genetic tools, we found that gap junctions are present in *Drosophila* Kenyon cells (KCs), the major neurons of the mushroom bodies (MBs), and showed that they play an important role in visual learning and memory. Using a dye coupling approach, we determined the distribution of gap junctions in KCs. Furthermore, we identified a single pair of MB output neurons (MBONs) that possess a gap junction connection to KCs, and provide strong evidence that this connection is also required for visual learning and memory. Together, our results reveal gap junction networks in KCs and the KC-MBON circuit, and bring new insight into the synaptic network underlying fly's visual learning and memory.

**\*For correspondence:** yanli42@126.com (YL); akguo@ion.ac.cn (AG)

[†]These authors contributed equally to this work

**Competing interests:** The authors declare that no competing interests exist.

## Introduction

In addition to chemical synapses, neurons communicate via gap junctions, also called electrical synapses. A gap junction consists of an intercellular channel that is formed by two hemichannels, each containing six homo- or heteromeric protein subunits. These subunits belong to the *connexin* or *pannexin* gene families in vertebrates and the *innexin* (*inx*) gene family in invertebrates. Gap junctions allow the circulation of small molecules, such as ions and cAMP (*Hervé and Derangeon, 2013*; *Phelan and Starich, 2001*). The function of gap junctions in noise reduction (*DeVries et al., 2002*; *Veruki and Hartveit, 2002*) and neuronal synchronization (*Bloomfield and Völgyi, 2009*; *Pereda et al., 2013*) is critical for visual and olfactory processing in mammals. In addition, the synchronization of neurons through the function of gap junctions was proposed to play a key role in cognitive processes, such as perception, attention, as well as learning and memory (*Bissiere et al., 2011*; *Hormuzdi et al., 2004*; *Tamás et al., 2000*). The complexity of the mammalian central nervous system makes it difficult to determine the function of gap junctions during cognitive processes. In comparison, insects have much simpler neural systems, yet still perform sophisticated cognitive functions, however, the role of gap junctions in cognition has rarely been investigated.

There are 8 types of *inx* genes expressed in *Drosophila*. In the brain, it has been reported that *ogre* (*inx1*), *inx2* and *inx3* were expressed in interstitial glia and lamina with similar expression patterns; the expression of *inx5* and *inx7* were detected in the cell body rind with very similar domains, and *inx6* was detected uniquely in an outer layer of cell body rind; *inx6* and *inx7* were also discovered between the anterior paired lateral (APL) neuron and dorsal paired medial (DPM) neuron; *shakB*

(*inx8*) was expressed in scattered neurons, giant fiber system and antenna lobe (*Stebbings et al., 2002*; *Wu et al., 2011*; *Yaksi and Wilson, 2010*). Gap junctions in giant fiber system are necessary for the escape behavior (*Phelan et al., 1996*). Gap junctions connect L1 with L2 neurons in the lamina, allowing for simultaneous activation of both pathways (*Joesch et al., 2010*). Gap junctions in olfactory glomeruli mediate lateral excitation which promotes olfaction sensitivity (*Yaksi and Wilson, 2010*). Especially, heterotypic gap junctions between APL and DPM neuron are critical for anesthesia-sensitive memory (*Wu et al., 2011*).

Mushroom bodies (MBs) are a pair of clearly distinguishable structures in the *Drosophila* brain, and play critical roles in cognition, such as olfactory coding (*Lei et al., 2013*; *Masse et al., 2009*), olfactory learning and memory (*Davis, 2011*; *Wolf et al., 1998*),context generalization (*Liu et al., 1999*), feature extraction (*Peng et al., 2007*), reversal learning (*Ren et al., 2012*; *Wu et al., 2012*), and value based decision making (*Tang and Guo, 2001*; *Zhang et al., 2007*). The precise role of MBs in visual learning and memory, however, remains under debate. It was thought previously that MBs are not necessary for visual operant conditioning in a flight simulator system (*Wolf et al., 1998*); however, recently it was reported that the MB γ lobe participates in visual learning and memory in color-based visual learning assays in groups of walking flies (*Vogt et al., 2014*). Kenyon cells (KCs), the major neurons of MBs, are the main components of the MB neural network, which is composed of dopamine neurons (DAs), MB output neurons (MBONs), and other cells (*Aso et al., 2014a, b*; *Tanaka et al., 2008*). While eight *inx* genes are expressed in *Drosophila*, the existence of functional gap junctions in KCs remains to be investigated. Here, we set out to study the presence and function of gap junctions in MBs, using the visual conditioning paradigm in a flight simulator as well as dye coupling and whole-cell recording approach.

## Results

### Non-chemical-synaptic transmission in MBs participates in visual learning and memory

To study the role of MBs in visual cognition in flying flies, we combined an optogenetic stimulator with a flight simulator system. A laser was focused to an area between the two compound eyes of the fly to activate halorhodopsin (eNpHR3.0) (*Gradinaru et al., 2010*), and *all-trans* retinal (ATR) was replaced with β-carotene to avoid photolysis (*Figure 1A*, *Figure 1—figure supplement 1A–D*). We found that visual learning and memory was impaired when α/β neurons of KCs, driven by 17d-GAL4, were hyperpolarized, and the memory was restored when GAL4 activity was blocked in MBs by MB247-GAL80 (*Figure 1B*, upper panel). These experiments were repeated with a second UAS-NpHR strain, yielding similar results (*Figure 1—figure supplement 1E*). Hyperpolarizing α'/β' neurons with MB370B-GAL4, γ neurons with MB131B-GAL4, or α/β and γ neurons of KCs with MB247-GAL4 also abolished visual learning and memory (*Figure 1B* lower panel). These results indicate that KCs participate in visual learning and memory. However, when synaptic release was impaired via expression of *shibire^{ts1}* (*shi^{ts1}*) (*Kasuya et al., 2009*; *Praefcke and McMahon, 2004*; *van der Bliek and Meyerowitz, 1991*) in KCs by 17d-GAL4, MB370B-GAL4 or MB247-GAL4, visual learning and memory was intact (*Figure 1C*). Tetanus toxin (TNT) was also used to inhibit the chemical-synaptic transmission. When TNT was expressed in α/β neurons (17d-GAL4) or all KCs (VT61721-GAL4), fly's visual learning and memory remained intact (*Figure 1—figure supplement 2*). In addition to blocking the output of chemical synapses, hyperpolarization could also affect other mechanisms that influence visual learning and memory. To confirm the effect of hyperpolarization in KCs, Kir2.1, ectopic expression of an inward rectifier potassium channel that induces irreversible hyperpolarization was achieved in KCs by using the MB247–GeneSwitch GAL4 driver (MBSwitch) (*Osterwalder et al., 2001*). As shown in *Figure 1D*, visual learning and memory were again abolished. However, in agreement with previous report (*Wolf et al., 1998*), visual learning and memory remained intact in flies of which MBs were ablated by hydroxyurea (HU) (*Figure 1—figure supplement 3*). Taking together, these results indicated that visual learning and memory might be influenced in real time by non-chemical-synaptic transmission in MBs. We therefore speculated that electrical-synaptic transmission via gap junctions was a plausible candidate for such observation.

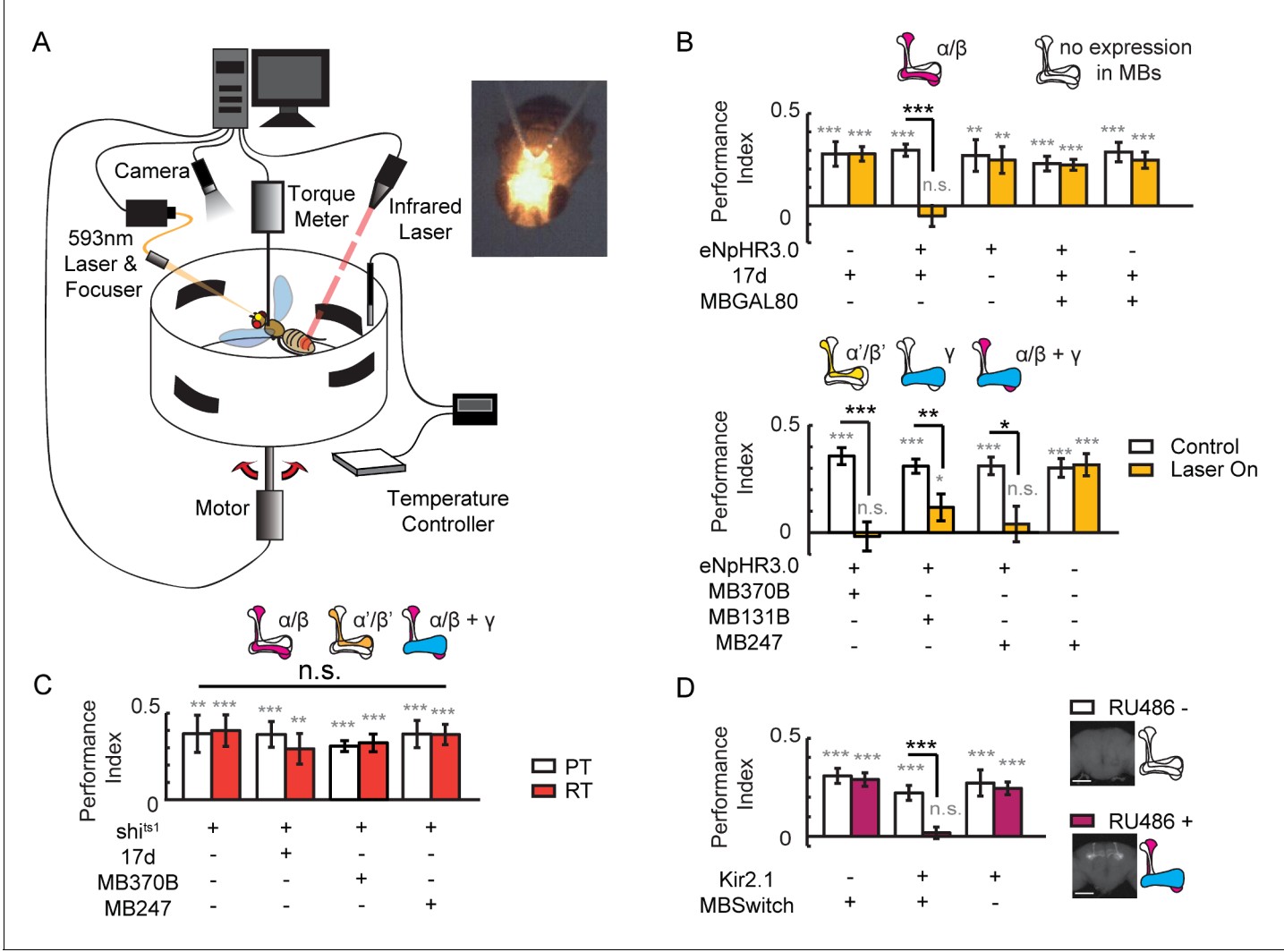

**Figure 1.** Visual learning and memory is abolished upon hyperpolarization of KCs. (A) Setup of the optogenetic flight simulator system. A 593 nm laser was used to activate halorhodopsin. To avoid influencing the visual perception of the flies, a focuser was used to focus the laser into a Gaussian light spot and to direct it in between the two compound eyes (diameter, ~0.3 mm). (B) Visual learning and memory was impaired when UAS-eNpHR3.0-YFP was used to hyperpolarize KCs labeled with 17d-GAL4, MB370B-GAL4, MB131B-GAL4 and MB247-GAL4. The expression pattern of these GAL4 lines are illustrated above the bars. MB247-GAL80 restored memory impaired by17d-GAL4. The laser was off in the control groups (Control) and applied throughout the 24min assay (at 0.1 mW/mm²) in the test groups (Laser On). (C) Blocking the chemical synaptic output of KCs labeled with 17d-GAL4, MB370B-GAL4 and MB247-GAL4 did not affect visual learning and memory. The temperature was unchanged throughout the 24 min assay. The permissive temperature (PT) was 24 ± 1°C, and the restrictive temperature (RT) was 30 ± 0.5°C. (D) Hyperpolarizing KCs labeled with MB247-GeneSwitch impaired visual learning and memory. For 3 days prior to start of the experiment, the flies in the "RU486+" group were fed with RU486 (50 mM dissolved in EtOH, 1:100 mixed with food), and the flies in the "RU486–" group were fed with 1% EtOH in food. Scale bar, 100 μm. The expression pattern of each driver is illustrated. N=15 to 20 for each data point. All data represent the mean ± SEM. Asterisks in grey indicate the level of statistical significance of the performance indices compared against chance level. Asterisks in black indicate the level of statistical significance of the performance indices between groups. (*p<0.05; **p<0.01; ***p<0.001).

The following figure supplements are available for figure 1:

**Figure supplement 1.** Establishing the optogenetic flight simulator system.

**Figure supplement 2.** Blocking the chemical synaptic output of KCs did not impair visual learning and memory.

**Figure supplement 3.** HU-ablation of MBs did not impair visual learning and memory.

## Gap junctions in MBs participate in visual learning and memory

To test this hypothesis, we individually knocked down all eight *inx* genes individually in KCs using a RNAi approach (*Wu et al., 2011*). The efficiency of all RNAi was examined, and the resulting knockdown was at least 50% (*Figure 2—figure supplement 1*). The specificity and target sequences of RNAi are shown in *Supplementary file 1*. As shown in *Figure 2A*, knockdown of *inx6* in 17d-GAL4-labeled neurons resulted in impairment of fly learning and memory. Flies with the knockdown of *inx4*, *inx5* or *inx6* were not able to achieve performance indices significantly different to zero, whereas the control groups did. The effect of gene downregulation was eliminated when the RNAi expression were blocked by MBGAL80. Visual learning and memory was also impaired in flies with the knockdown of *inx4*, *inx5* or *inx6* in MB247-GAL4-labeled KCs (*Figure 2B*). To exclude potential developmental effects, MBSwitch was used to induce RNAi expression in adult stage only. When *inx4* was knocked down in adults, visual learning and memory was not affected, whereas adult knockdown of *inx5* or *inx6* exerted significant effects (*Figure 2C*). Similar results were obtained in experiments using independent RNAi strains (*Figure 2D*, JF5 target to inx5 and JF6 to inx6). Therefore, INX5 and INX6 are post-developmentally required in the $\alpha/\beta$ lobes of MBs for visual learning and memory whereas INX4 may contribute to MB development instead, thus affecting visual learning in an indirect manner.

We further tested whether gap junctions in other KC neurons also participate in visual learning and memory. The split-GAL4s MB370B and MB131B were used to drive *inx*-RNAi expression in $\alpha'/\beta'$ neurons or $\gamma$ neurons, respectively. As shown in *Figure 2E*, knockdown of *inx5* or *inx6* in $\alpha'/\beta'$ neurons abolished visual learning and memory, and knockdown of *inx* genes in $\gamma$ neurons had no effect (*Figure 2F*). Together, these results suggest that the gap junctions composed of INX5/6 in $\alpha/\beta$ and $\alpha'/\beta'$ neurons are involved in visual learning and memory in *Drosophila*.

## Gap junctions couple KCs of the same subtype or different subtypes

To verify the existence of gap junctions in KCs, we utilized a dye coupling method. Either biocytin or Alexa Fluor 568 was loaded into one KC of each fly brain upon whole-cell recording. The dye-filled neurons were identified based on their position and morphological characteristics as previously reported (*Aso et al., 2014a*; *Lee et al., 1999*). Among the 143 successfully stained brains, 64 $\alpha/\beta$ neurons, 40 $\alpha'/\beta'$ neurons and 39 $\gamma$ neurons were loaded with dye. Dye-coupled neurons were observed in 14 brains (*Figure 3A*). In 10 out of the 14 brains, dye coupling was observed between the same subtype of KCs, including 8 $\alpha/\beta$-$\alpha/\beta$, 1 $\alpha'/\beta'$-$\alpha'/\beta'$ and 1 $\gamma$-$\gamma$ (*Figure 3A*). One representative example is shown in *Figure 3* and *Video 1*. There, biocytin was loaded into a GFP-expressing $\alpha/\beta$ posterior ($\alpha/\beta$ p) neuron (*Tanaka et al., 2008*) (driven by c205-GAL4). Across the calyx, the MB dendrites region, there was only one biocytin-filled fiber (*Figure 3B*); however, two biocytin-filled fibers were found in the peduncle and the lobe region (*Figure 3C*). One fiber overlapped with the GFP fluorescence, while the other fiber showed no signs of GFP expression (*Figure 3D*). Tracing the two fibers, we found that one of them belongs to an $\alpha/\beta$ p neuron, and the second belongs to an $\alpha/\beta$ core ($\alpha/\beta$ c) neuron (*Figure 3D,E*). Notably, there were three potential connection points between these two neurons, including the beginning of the lobes, the horizontal branch tip of the $\alpha/\beta$ p neuron, and the middle of the vertical branches (*Figure 3D,E* right panel, marked with "*"). In 4 of the 14 brains, dye coupling was observed between different KC subtypes (*Figure 3A*). In 2 out of the 4 brains, the dye was loaded into a single $\gamma$ neuron but stained multiple $\gamma$ neurons and an $\alpha'/\beta'$ neuron (*Figure 3—figure supplement 1* and *Video 2*). In the third brain, one $\alpha'/\beta'$ soma was filled with dye and two $\alpha'/\beta'$ somas, one $\alpha/\beta$ and one $\gamma$ fiber were stained (*Figure 3—figure supplement 2* and *Video 3*). In the last brain, the dye was loaded into a single $\alpha/\beta$ p neuron, and was also found in multiple $\alpha'/\beta'$ neurons and some non-KC neurons (*Figure 3—figure supplement 3* and *Video 4*). Together, these results suggest that gap junctions exist in both the calyx and the lobes that link KCs of the same or different subtypes as well as some non-KC neurons.

To rule out false-positive results in the dye coupling experiments, blind experiments were performed in flies with *inx5/6* double knockdown in all KCs (driven by VT61721-GAL4) and wild type flies. Ninety brains were stained: 38 from wild type flies and 52 from *inx5/6* knock down flies. Dye coupling was observed in 5 brains of the wild type flies but in none of the *inx5/6* knockdown flies (*Table 1*). Assuming a binomial process with p=5/38 (k = 5; n = 38 in the control group) to observe

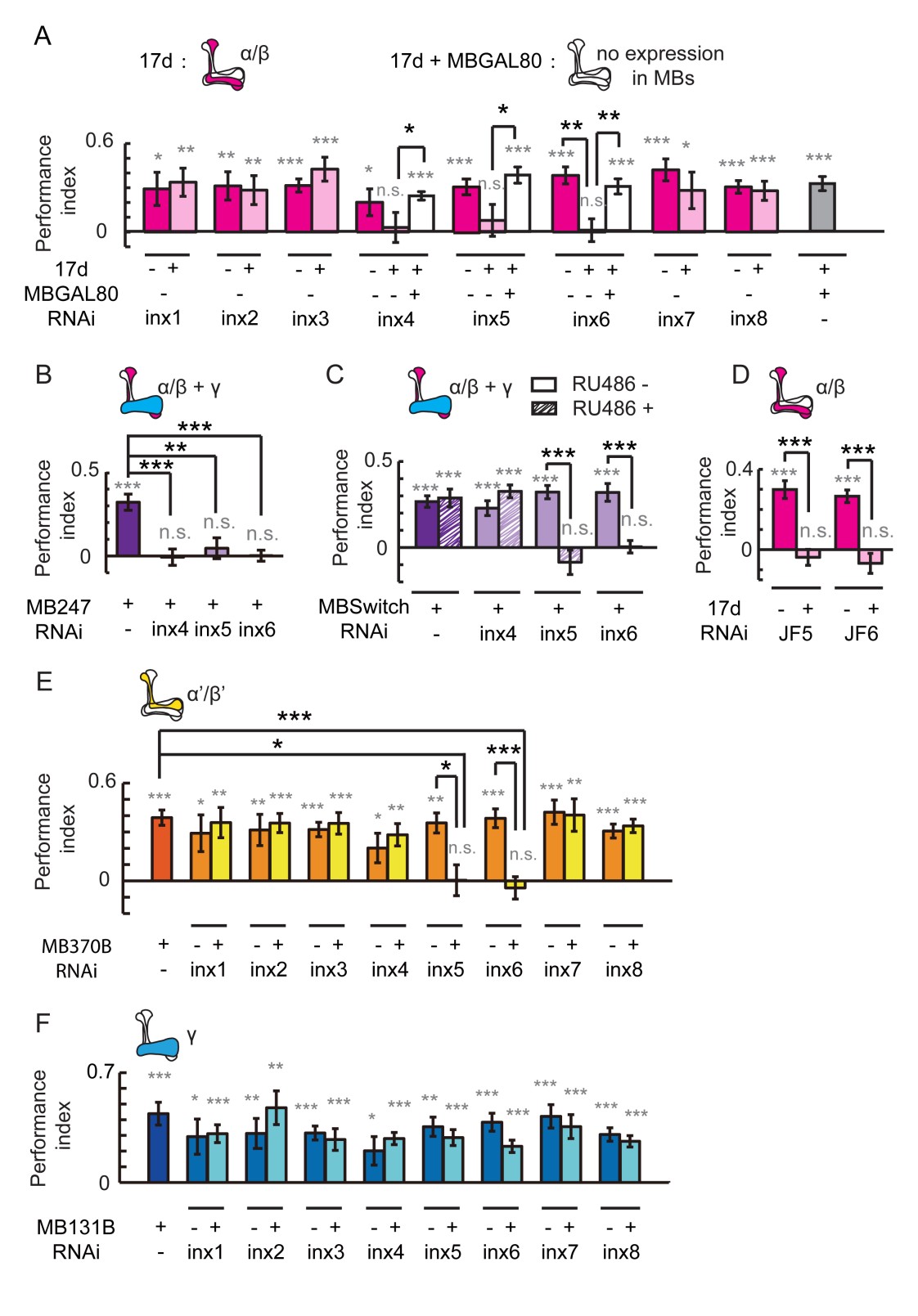

**Figure 2.** Gap junctions in MBs are necessary for visual learning and memory. (**A**) Knockdown of *inx6* in KCs labeled with 17d-GAL4 impaired visual learning and memory. Knockdown of *inx4* or *inx5* eliminate the significance of the difference between the performance indices and zero. The effect of *inx4, inx5* or *inx6* knockdown was rescued by MBGAL80. (**B, C**) Knocking down *inx4, inx5* or *inx6* in KCs from an early development stage impaired visual learning and memory (**B**), while knockdown of *inx4* in KCs after eclosion showed no effect (**C**). For 3 days before the experiment, the flies in the

*Figure 2 continued on next page*

*Figure 2 continued*

RU486+ group were fed with RU486 (50 mM stock dissolved in EtOH, 1:100 mixed with food), and the flies in the RU486– group were fed with 1% EtOH in food. (**D**) Knockdown of *inx5* or *inx6* in KCs labeled with 17d-GAL4 with another type of RNAi also impaired visual learning and memory. (**E**) Knockdown of *inx5* or *inx6* in MB370B split-GAL4-labeled α'/β' neurons impaired visual learning and memory. (**F**) Knockdown of each of the *inx* genes in MB131B split-GAL4-labeled γ neurons had no effect on visual learning and memory. The expression pattern of each driver is illustrated. The experiments in (**C, D**) were performed as blind experiments. The following RNAi lines were used: *inx1*: v103816; *inx2*: v102194; *inx3*: v39094; *inx4*: v33277; *inx5*: v102814; *inx6*: v8638; *inx7*: v103256; *inx8*: v26801; JF5: TRiP. JF02877; and JF6: TRiP. JF02168. The specificity and target sequences of RNAi are shown in *Supplementary file 1*. N = 15 to 20 for each data point. All data represent the mean ± SEM. Asterisks in grey indicate the level of statistical significance of the performance indices compared against chance level. Asterisks in black indicate the level of statistical significance of the performance indices between groups. (*p<0.05; **p<0.01; ***p<0.001).

The following figure supplement is available for figure 2:

**Figure supplement 1.** Effectiveness of *inx* RNAi lines.

false-positive dye coupling, the probability of failing to observe any coupling in the knockdown group (k = 0; n = 52) is about 0.00065, suggesting a small likelihood for false-positive labeling.

Data are from blind experiments. 38 wild type (WT) and 52 *inx5/6* knockdown (KD) brains were loaded with biocytin. The numbers of brains for each condition are shown.

## Gap junctions between MBs and MBONs participate in visual learning and memory

In our dye coupling experiments, we found some dye-stained fibers from the area of β' lobe to areas outside the MBs and two dye-stained non-KC somas (*Figure 3—figure supplement 3* and *Video 4*). Based on the morphology of these fibers and the position of the somas, these non-KC neurons were likely to be MBONs. MBON-β'2mp neurons which have somas on the dorsal anterior surface near the mid-line of the brain, and arborize on the tip of β' lobe, were supposed to be reasonable candidates. To investigate the possible existence of an efferent pathway mediated by gap junctions in MBs, we explored whether there were electrical synapses between MBONs and KCs. We used either the proton pump eArch3.0 (*Mattis et al., 2012*) or halorhodopsin (*Gradinaru et al., 2010*) to hyperpolarize KCs (247LexA > LexAop-eArch3.0-EYFP or 247LexA > LexAop-eNpHR3.0-EYFP), and MBONs were labeled with split GAL4-driven tdTomato (*Figure 4—figure supplement 1*). Using whole-cell recording, we monitored the changes in membrane potential in MBONs whilst KCs were hyperpolarized simultaneously using a yellow laser (*Figure 4A*). Considering that the electrical coupling coefficient might not be high (*Yaksi and Wilson, 2010*), we used the maximal power of our laser, which should be the most promising condition to get obvious results. Inasmuch as there was no significant difference between the effect of eArch3.0 and eNpHR3.0 to hyperpolarize KCs (*Figure 4—figure supplement 1C*), the data using eArch3.0 or eNpHR3.0 were pooled together for analysis. Upon hyperpolarization in KCs, we measured a hyperpolarization in MBON-β'2mp neurons labelled by MB002B-GAL4 or MB011B-GAL4. This response remained unchanged after the application of tetrodotoxin (TTX), which blocks chemical synapses by suppressing action potentials (*Figure 4B*). In contrast, laser-induced hyperpolarization disappeared when gap junctions were blocked by 2-octanol (2-OCT) (*Bohrmann and Haas-Assenbaum, 1993*), and was recovered after 2-OCT was washed out (*Figure 4C*), while 2-OCT application could induce slight hyperpolarization (*Figure 4—figure supplement 2*). The delivery of 2-OCT did not affect the optogenetic hyperpolarization in KCs (*Figure 4—figure supplement 1*). These results suggest that β'2mp MBONs are able to receive inputs from KCs through electrical synapses.

During whole-cell recording, biocytin was loaded into the recorded neuron, therefore, the identity of this recorded neuron could be subsequently confirmed (*Figure 4A* and Materials and methods). In 3 of the 35 recorded brains, dye coupling was observed between a β'2mp neuron and an α'/β' neuron. One example is shown in *Figure 5*, *Video 5* and *6*. In addition to the arborization of the β'2mp neuron, biocytin also stained neurites in the β' lobe (*Figure 5A,B*), vertical lobe stalk (*Figure 5A–C*), tip of the α' lobe (*Figure 5A,D*), peduncle (*Figure 5A, D, E*) and calyx (*Figure 5E*), indicating the presence of gap junctions between β'2mp neurons and α'/β' neurons (*Figure 5F*).

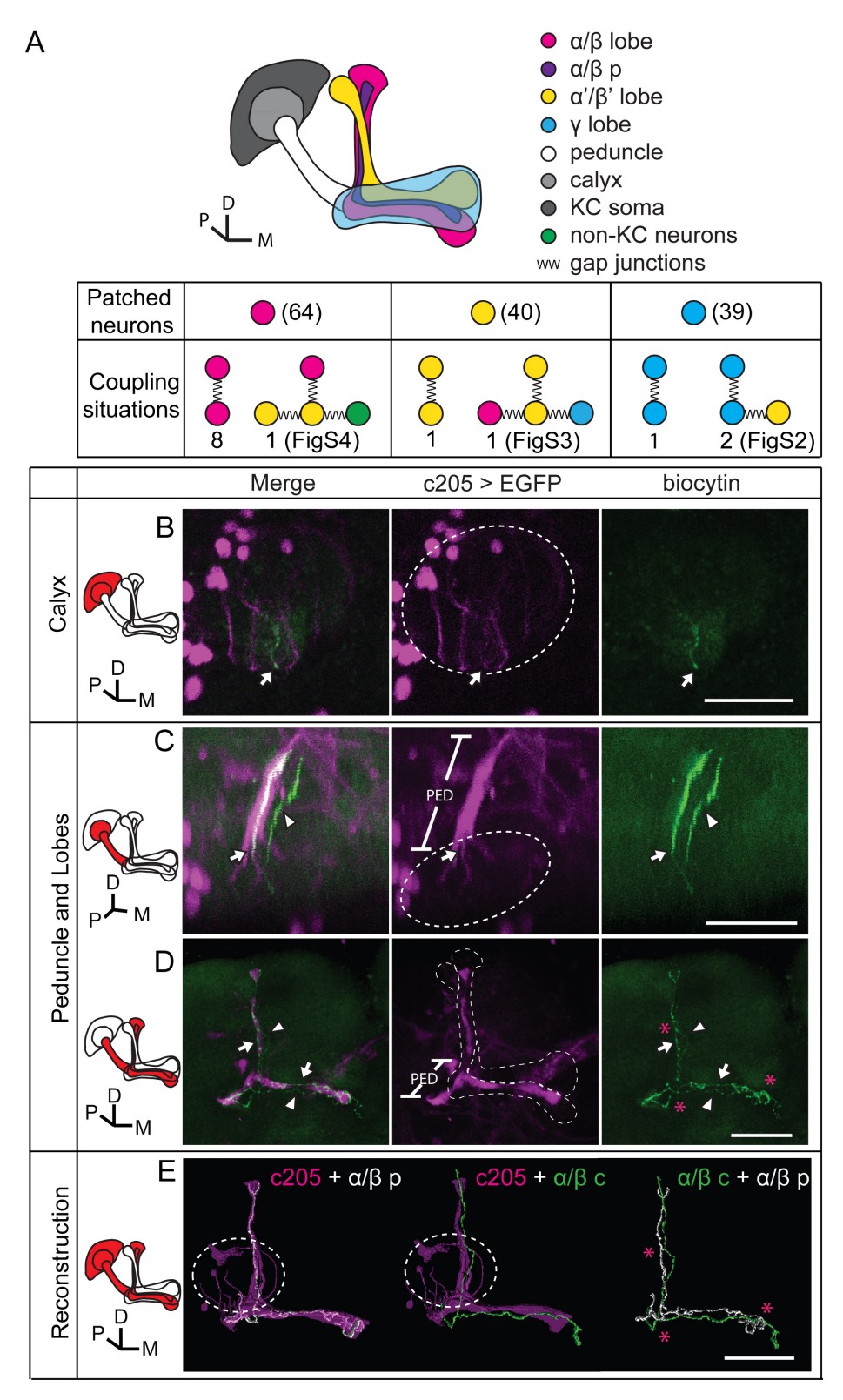

**Figure 3.** Dye coupling reveals presence of gap junctions in KCs. (A) Schematic representation of MB and summary of dye coupling results. (B–D) One example in which biocytin was loaded into one KC, labeling more than one KC. Magenta, fluorescence of UAS-mCD8::EGFP (EGFP)-labeled α/β p stratum (driven by c205-GAL4); green, biocytin-filled neurites. (B) Z stack of the calyx and somas. One biocytin-filled fiber was observed. Arrow, the biocytin-filled fiber of the patched neuron; dotted circle, the boundaries of the calyx. (C) Z stack of the peduncle and lobes. Arrowhead, second
*Figure 3 continued on next page*

*Figure 3 continued*

biocytin-filled fiber. PED, peduncle. (**D**) Z stack of the peduncle and lobes in MBs. "*", points to the overlap between the two biocytin–filled fibers. Dotted lines show the boundaries of the α/β lobe and the α'/β' lobe. (**E**) Reconstructed morphology of the MB. Left, one neurite (white) was coincident with the α/β p stratum and presumed to be an α/β p neurite; middle, the other neurite (green) was not coincident with the α/β p stratum and was presumed to be an α/β core (α/β c) neurite; Right, the 2 neurites. D, dorsal; M, medial; P, posterior. Scale bar, 30 μm.

The following figure supplements are available for figure 3:

**Figure supplement 1.** Dye coupling between two γ neurons and one α'/β' neuron.

**Figure supplement 2.** Dye coupling between α/β, α'/β' and γ neurons.

**Figure supplement 3.** Dye coupling between α/β, α'/β' and non-KC fibers.

Next, we investigated whether the gap junctions in β'2mp neurons participate in visual learning and memory. Each of the 8 *inx* genes were knocked down in β'2mp neurons individually using the split-GAL4 MB002B. As shown in *Figure 6*, visual learning and memory were ablated following *inx6* knockdown using two set of RNAi strains from independent sources. Downregulation of *inx5* with the RNAi strain TRiP. JF02877 also abolished visual learning and memory (*Figure 6*). Taken together, our results provide strong evidence that a gap junction network containing KCs and β'2mp neurons is crucially involved in visual learning and memory.

## Discussion

The function of the MBs, has now been studied for several decades. However, a possible involvement of gap junctions in cognition had not been well addressed, especially in KCs. Here, we showed that in addition to chemical synapses, gap junctions also exist in KCs and KC-associated neural networks, and play important roles in visual learning and memory. Compared with chemical synapses, gap junctions are simple, fast, and non-discrete for information transfer in neural networks. Similar to their function as a low-pass filter in reducing noise during sensory processing (*DeVries et al., 2002*; *Veruki and Hartveit, 2002*; *Zhang et al., 2013*), gap junctions in KCs may contribute to noise reduction, especially those that connect KCs of the same subtype with common inputs. Considering that the different KC subtypes have diverse functions (*Guven-Ozkan and Davis, 2014*; *Vogt et al., 2014*; *Yi et al., 2013*), gap junctions that connect KCs of different subtypes may serve as a communication channel that links multiple processing units. The gating, assembly, internalization and degradation of gap junction channels can be delicately modulated by phosphorylation (*Laird, 2005*; *Moreno, 2005*; *Solan and Lampe, 2005*); and these regulation may underlie memory encoding in MB-MBON networks.

Our results showed that approximately 10% of all KCs possess active gap junction connections, which is very likely to be an underestimate of their density, due to the following reasons: 1) The dye coupling efficiency may be restricted with the gating of gap junction channels, which might not be always open and which might be delicately modulated; 2) filling an entire neuron with sufficient amounts of dye is a difficult and time-consuming (1–2 hr) procedure, due to the slender morphology of their neurites. According to electron microscope results from our lab (unpublished data), KC neurites are extremely narrow in various sections, often only 50 nm in diameter. However, some organelles in the neurites, such as mitochondria, are quite big, with the diameter more than 500 nm. Thus,

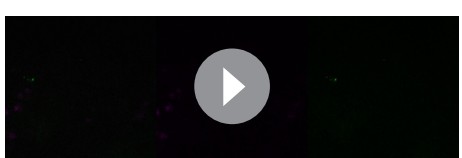

**Video 1.** Raw stack of the brain in *Figure 3*.

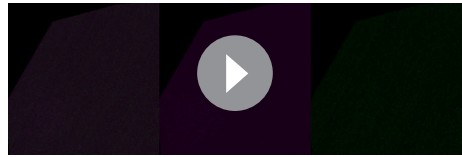

**Video 2.** Rotated stack of the brain in *Figure 3—figure supplement 1*.

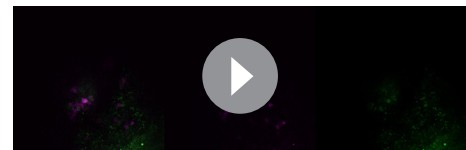

**Video 3.** Raw stack of the brain in *Figure 3—figure supplement 2*.

**Video 4.** Raw stack of the brain in *Figure 3—figure supplement 3*.

the diffusion of dye in KCs might be interrupted when neurites were congested by organelles. In general, Alexa Fluor 568 (791.8 Da) stained less details of the neurites than biocytin (372.48 Da) in KCs (*Figure 3—figure supplements 1* and *2*). This difference also suggested that the KC neurites are not free enough for the dye diffusion. It is therefore unsurprising that the neurites of the coupled neurons were rarely completely filled with dye, with similar reasoning for the soma parts. The low efficiency of dye coupling was also evident in our results from $\beta'$2mp neurons. In this study, the probability of dye coupling was approximately 1/12 the probability of electrical coupling, which we believe is reasonable, as electrical currents flow more easily than dyes, both in neurites and through gap junctions. We could not exclude the possibility that the distribution of gap junctions between KCs, and between KCs and MBONs is rare, and possibly differs substantially across individual flies. It is worth noting that the proportion of simultaneously opened gap junctions is probably not very high, thus synchronizing only a minority of KCs. Otherwise, it would be akin to creating a single processing unit, and the MBs would lose their processing capacity. Thus, such fine-tuned modulation of gap junctions may be critical for MB function.

Our findings indicate that INX 5 and 6 in KCs and MBON-$\beta'$2mp neurons are necessary for visual learning and memory. Nevertheless, the features of gap junctions involved are not yet determined, such as the constitution (homotypic or heterotypic), the conductance and the rectification properties of particular gap junction channels in particular neural networks. Combining whole-cell recording with both depolarizing and hyperpolarizing optogenetic tools could be helpful to study the electrical physiological properties of gap junctions, and further clarify the dynamics of neural networks with both chemical synapses and gap junctions.

Our results suggest that MBs take part in visual learning and memory via gap junctions. However, in previous reports, visual learning remained intact in the *mbm*[1] mutant and MBs-ablated flies (*Liu et al., 1999*; *Wolf et al., 1998*). The experiment was repeated with HU-treated MBs-ablated flies in our flight simulator, yielding similar results (*Figure 1—figure supplement 3*). Therefore, this unconformity is unlikely to be caused by the upgrading of the flight simulator. Developmental compensation may be a reasonable explanation for this discrepancy in results. HU inhibits ribonucleotide reductase, blocking the synthesis of DNA, and causing the death of dividing cells (*Sweeney et al., 2012*). It has been reported that the trajectory of the neurons which proliferated during larva period develop independently of each other, although the final branching pattern of these neurons is dependent upon the presence of appropriate neuronal targets (*Lovick and Hartenstein, 2015*). In the MB ablated flies, MB-targeting neurons might still grow to the MB region and form new circuits with the remainder γ neurons which proliferated before larval hatching and therefore could not be ablated by HU (*Figure 1—figure supplement 2C*). The developmental compensation might alleviate the change of the topological structure and function of the MB-associated neural network. A comparison of the MB-associated network between the MB-ablated flies and wild type flies, including

**Table 1.** Dye coupling in wild type and *inx5/6* knockdown flies.

|  |  | WT | INX 5&6 KD |
|---|---|---|---|
| Loaded neurons | αβ | 21 | 27 |
|  | α′β′ | 9 | 10 |
|  | γ | 8 | 15 |
| Dye coupling |  | 5 | 0 |

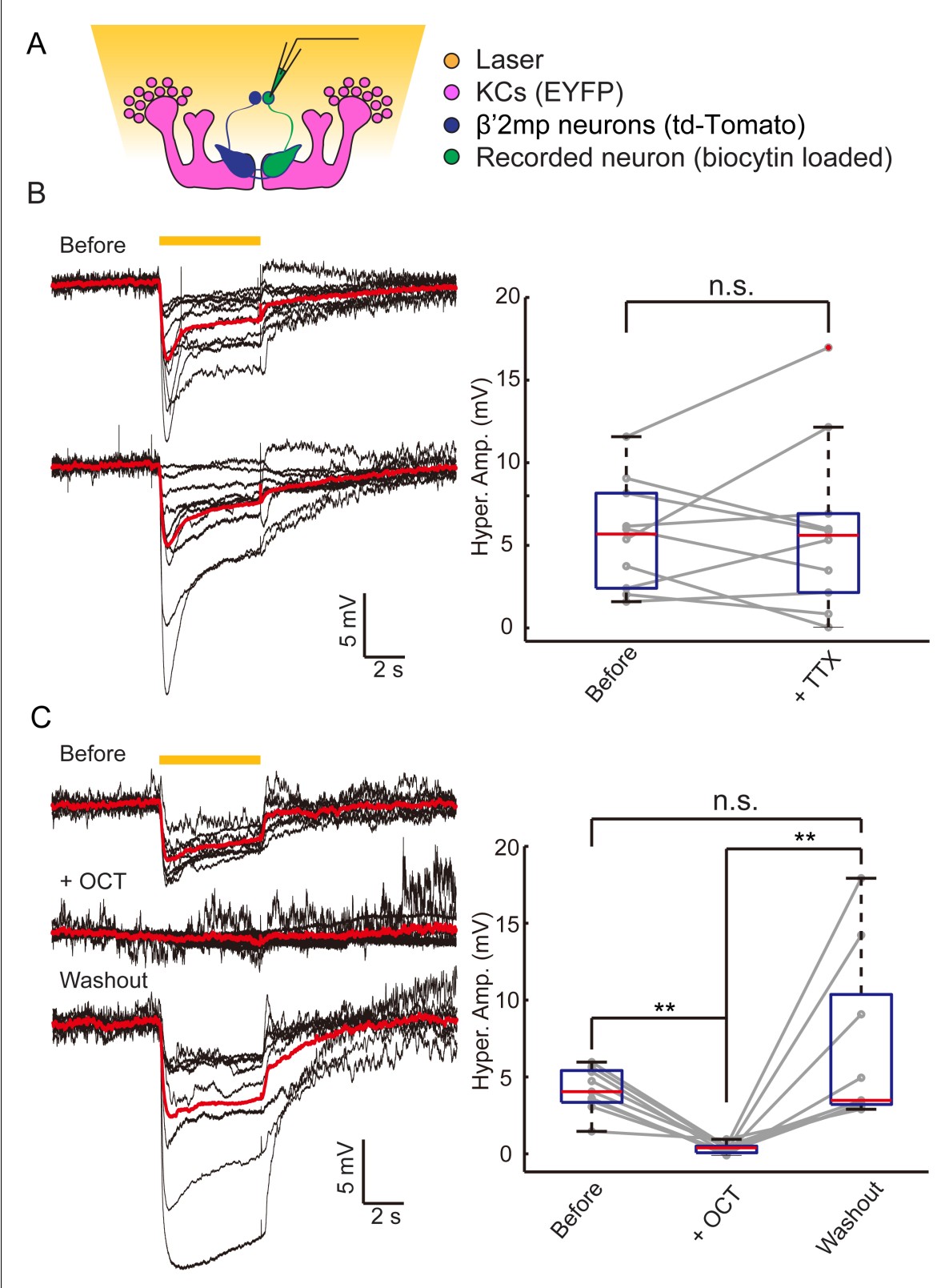

**Figure 4.** Electrical coupling reveals presence of gap junctions between KCs and β'2mp neurons. (**A**) Strategy used to detect electrical coupling between MBONs and KCs. Membrane potentials of MBON somata were recorded during hyperpolarization of KCs using optogenetics. The recorded soma was loaded with biocytin for later identification. (**B**) Whole-cell recordings were performed on MBON-β'2mp neurons while KCs were hyperpolarized using laser light with emission at 593 nm (yellow bar). Hyperpolarization was observed in the recorded β'2mp neurons. TTX delivery did

*Figure 4 continued on next page*

*Figure 4 continued*

not affect hyperpolarization. Black traces, average of the membrane-potential traces for each recorded $\beta'2mp$ neurons (n = 10). Red traces, average of the black traces. Box plots show the peaks of the hyperpolarization amplitudes (Hyper. Amp.) before and after TTX delivery. (**C**) MBON $\beta'2mp$ neurons were recorded as in (**B**). Hyperpolarization was eliminated after the application of the gap junction blocker 2-OCT and recovered after washout of 2-OCT (n=9). Box plots show the hyperpolarization amplitudes. Performance indices represent the mean ± SEM.

The following figure supplements are available for figure 4:

**Figure supplement 1.** No difference observed between NpHR- and Arch-mediated hyperpolarization of KCs.

**Figure supplement 2.** Membrane-potential traces of MBON-$\beta'2mp$ neurons in the OCT experiments.

the topological structure and the physiological responses during visual operant conditioning, would be helpful to understand the neural computations underlying visual cognition.

Recently, it was reported that blocking the output of MB $\gamma$ lobes with $shi^{ts}$ affects visual learning and memory when assessed using two color-based, multiple walking fly participation paradigms (*Vogt et al., 2014*). However, in the experiments using a flight simulator, visual conditioning was not affected when the chemical synaptic output of the $\alpha/\beta$ and $\gamma$ lobes was inhibited by $shi^{ts}$ or TNT. This discrepancy can probably be attributed to the differences between the behavioral paradigms. In the paradigms for the walking flies (*Vogt et al., 2014*), to ensure the blocking of $shi^{ts}$, the flies was tested at high temperature (33°C), but in flight simulator, files could not accomplish the visual learning and memory task when the temperature was higher than 30°C. Moreover, the efficiency of TNT could not be tested in these behavior paradigms. Thus, we cannot formally rule out the possible roles of chemical transmission in visual learning and memory. Our results show that gap junctions in the $\alpha/\beta$ and $\alpha'/\beta'$ lobes, but not those in the $\gamma$ lobes, are required for the operant visual learning, suggesting that the chemical synaptic circuits and electrical synaptic circuits may be complementary in distribution and function. Therefore, we propose that gap junctions cooperate with chemical synapses to perform the complex function of MBs. Tracing the gap junction networks in *Drosophila* and clarifying the synergy between gap junctions and chemical synapses should contribute to our understanding of functional neural calculations.

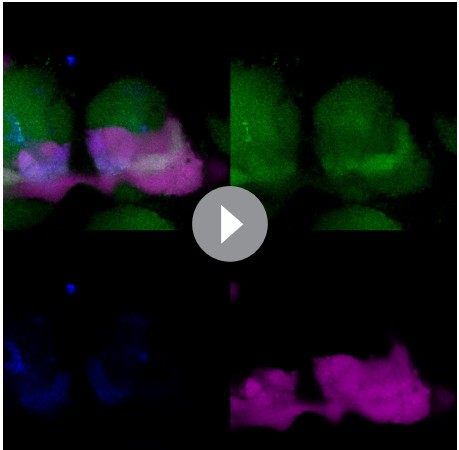

**Video 5.** Raw stack of the anterior half of the brain in *Figure 5*.

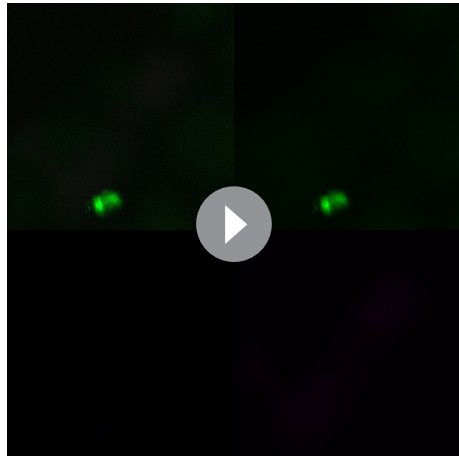

**Video 6.** Raw stack of the brain in *Figure 5*.

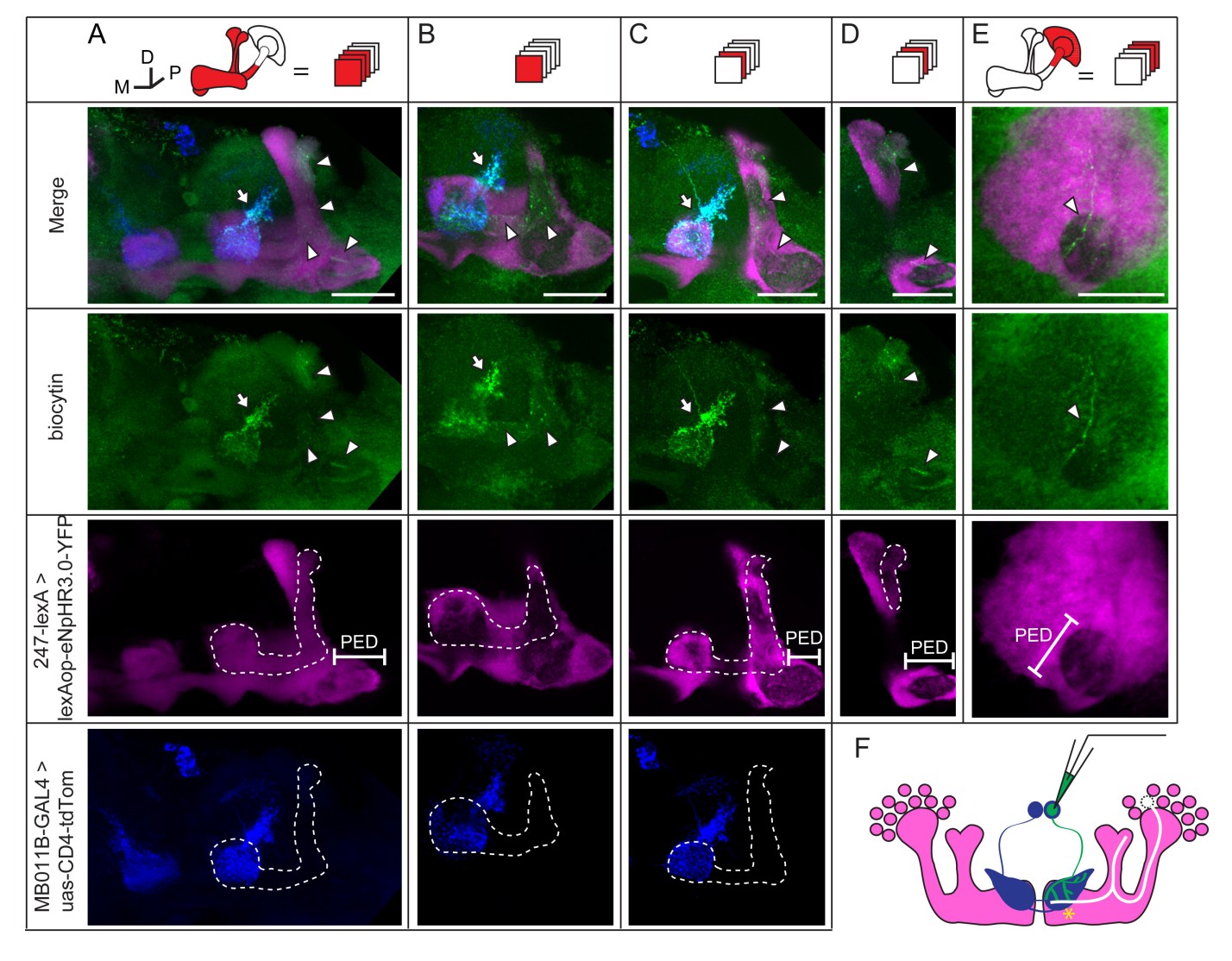

**Figure 5.** Dye coupling between KCs and β′2mp neurons. (A) Z stack of MB lobes and half of the peduncle. Biocytin-filled neurites (arrowhead) are observed within the boundary of the α′/β′ lobe (dotted line) and peduncle (PED). (B–D) The lobe layers were divided into 3 parts from anterior to posterior and then stacked. In each stack, biocytin-filled neurites are visible within the boundaries of theα′/β′ lobe and peduncle. (E) Z stack of KC somas, the calyx and half of the peduncle. One biocytin-filled fiber is seen Green, biocytin-loaded neurites; magenta, the fluorescence of YFP-labeled KCs (driven by MB247-lexA); blue, the fluorescence of td-Tomato-labeled MBONs (driven by MB011B-GAL4). Arrow, arborizations of β′2mp neurons projecting to the tip of the β′ lobe; arrowhead, the biocytin-filled KC fibers; dotted line, the boundary of the α′/β′ lobe. PED, peduncle. Scale bar, 30 μm. (F) Schematic diagram of the brain. White, the α′/β′ neuron that was dye-coupled with the β′2mp neurons. "*", the presumed points of gap junctions.

## Materials and methods

### Fly strains

Flies were reared in standard food vials at 25°C and 60% relative humidity with a 12 hr dark/light (L/D) cycle (*Guo et al., 1996*), excluding the flies that were indicated as raised in the dark and the flies crossed with UAS-shibire[ts1];;shibire[ts1], which were reared at 21°C. We constructed the UAS-NpHR-EYFP, LexAop-eNpHR3.0-EYFP and LexAop-eArch3.0-EYFP fly strains. The NpHR-EYFP plasmid was provided by Dr. Liping Wang (Shenzhen Institutes of Advanced Technology, Shenzhen, China). The NpHR-EYFP fragment was amplified by PCR, cloned into the pUAST vector, and randomly inserted into the genome of W[1118] flies. The eNpHR3.0-EYFP and eArch3.0-EYFP plasmids were obtained

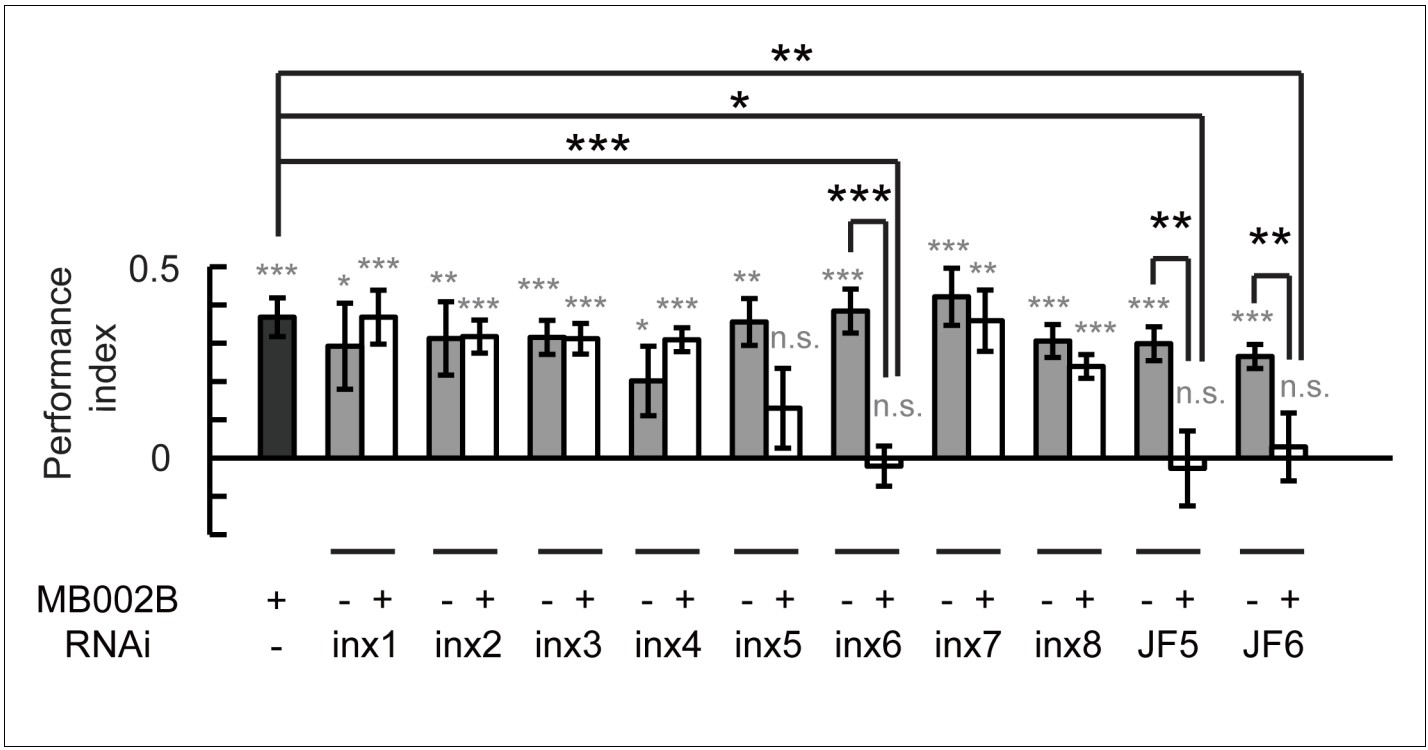

**Figure 6.** Gap junctions between KCs and *β'2mp* neurons are necessary for visual learning and memory. Knockdown of *inx5* or *inx6* in *β'2mp* neurons results in impaired visual learning and memory. The following RNAi lines were used: *inx1*: v103816; *inx2*: v102194; *inx3*: v39094; *inx4*: v33277; *inx5*: v102814; *inx6*: v8638; *inx7*: v103256; *inx8*: v26801; JF5: TRiP. JF02877; and JF6: TRiP. JF02168. N=17 to 20 for each data point. Asterisks in grey indicate the level of statistical significance of the performance indices compared against chance level. Asterisks in black indicate the level of statistical significance of the performance indices between groups. (*p<0.05; **p<0.01; ***p<0.001).

from Addgene. The eNpHR3.0-EYFP or eArch3.0-EYFP fragment was amplified, cloned into the pJFRC19 vector, and inserted into the 75B1 attP locus in vas-phi-zh2A-VK5flies, which were provided by Dr. Wei Wu (Institute of Biochemistry and Cell Biology, Shanghai, China). The 17d-GAL4, c205-GAL4, c305a-GAL4, c819-GAL4 and UAS-shibire[ts1];;shibire[ts1] fly strains were kindly provided by Dr. Li Liu (Institute of Biophysics, Beijing, China); and the MB247-GAL80 and MB247-lexA strains, by Dr. Scott Waddell (University of Oxford, Oxford, UK). The VT61721-GAL4 was a gift from Dr. Ann-Shyn Chiang (National Tsing Hua University, Hsinchu, Taiwan). The split-GAL4 lines MB002B, MB011B, MB131B and MB370B were obtained from the Janelia FlyLight Project Team and Dr. Gerald M. Rubin (Janelia Research Campus, Virginia). The *inx5* RNAi JF02877 and *inx6* RNAi JF02168 were provided by Dr. Luping Liu (Tsinghua University, Beijing, China), and the other *innexin* RNAi lines were obtained from the Vienna *Drosophila* RNAi Center. The specificity and target sequences of RNAi are shown in **Supplementary file 1**. The cha-GAL4, MB247-GAL4, OK107-GAL4, MB247-GeneSwitch, UAS-eNpHR3.0-EYFP, UAS-Kir2.1-GFP, UAS-CD4-tdTom and UAS-mCD8::EGFP were obtained from the Bloomington Stock Center. All other strains used in this study were extant lines from our lab. The 17d-GAL4, MB247-GAL4, c819-GAL4, c205-GAL4, MB247-GAL80, UAS-NpHR-EYFP, UAS-eNpHR3.0-EYFP and UAS-Kir2.1 fly strains were back-crossed with WCS flies.

## Optogenetics

In the optogenetic flight simulator system and the electrophysiological experiments, the yellow light (593 nm) used to activate halorhodopsin was generated using a semiconductor laser (Xi'an He Qi Opto – Electronic Technology Co., Ltd., China). The laser was focused into a Gaussian light spot (diameter: 0.3 mm with 99% of the power). For visual learning and memory experiments, the laser power was 0.1 mW/mm$^2$. Due to the possible low efficiency of electrical coupling, the laser power was maximized to 0.7 mW/mm$^2$ for electrophysiological experiments. Flies for visual conditioning were raised with a 12 hr L/D cycle and fed with food containing 100 μM *β*-carotene (C9750, Sigma-

Aldrich, St. Louis, Missouri) for 3~5 days before the experiments. For electrophysiological experiments, flies were raised in the dark and fed throughout their lives with food containing 100 µM ATR. In the optogenetic locomotor system, yellow light was supplied by LEDs (590 nm, 0.06–0.1 mW/mm2). ATR (R2500, Sigma-Aldrich, St. Louis, Missouri) or β-carotene was dissolved in absolute ethyl alcohol and mixed with food to a final concentration of 100 µM.

## Visual learning and memory paradigm

The flies were fixed with hooks to the clamp of a torque meter in the center of a circular panorama (44 mm diameter) that could be rotated by a fast electric motor (*Figure 1A*). The torque meter measured each fly's yaw torque around its vertical body axis and rotated the panorama around the fly via a negative feedback mechanism. This arrangement allowed the tethered flies to stabilize and choose their flight orientation with respect to the panorama by adjusting their yaw torque. The flies' yaw torque and the panorama's angular position were recorded continuously and stored in the computer (sampling rate, 20 Hz) for subsequent analysis. The panorama was divided into 4 quadrants, with horizontal bars at their respective centers. An infrared laser beam (10,600 nm) that projected to the fly's abdomen was switched on when the fly oriented to the dangerous quadrants and was switched off when the fly oriented to the safe quadrants. For half of the flies, the lower bars were coupled with punishment, and for the other half, the higher bars were coupled with punishment. Standard 24 min training paradigms consisted of 12 blocks of 2 min each: blocks 1–3, pre-training sessions; blocks 4, 5, 7 and 8, training sessions; blocks 6 and 9–12, test sessions. In our experiments, the horizontal bars were black on white backgrounds; the bars had a length of 40° and a width of 12°. The difference in the center of gravity between the higher and lower bars was 44°. Patterns in opposite quadrants were identical, whereas patterns in neighboring quadrants were different. These patterns were printed on paper with a color inkjet printer (Epson Stylus Photo 1390). The paper arena was illuminated by transmitted light from a white LED array. The performance of tethered flies was evaluated quantitatively in 2 min bins using the "performance index" (PI), which was calculated as follows: PI = (t1–t2)/(t1+t2), where t1 is the time spent heading toward safe patterns and t2 is the time spent heading toward dangerous patterns. The final PI was calculated as the average of PIs 9 to 12. The experiments in *Figure 2C and D* are performed as blind experiments, in which this paradigm was performed on flies by an experimenter who was unaware of the genotypes of the flies.

## MB ablation

We followed the procedure described previously (*de Belle and Heisenberg, 1994*). In brief, wild-type flies (3–7 d old) lay their eggs on agar plates containing sugar and grape juice for half an hour. The eggs were placed at 25°C for 18–20 hr, and newly hatched larvae (within 20 min) were quickly transferred into a yeast suspension containing 50 mg ml$^{-1}$ hydroxyurea (HU. H8627, Sigma) to ablate the MB neuroblasts. The yeast suspension for the vehicle group did not contain hydroxyurea. After 4 hr of treatment, larvae were washed and transferred into fresh food vials.

## Locomotor assay

Female flies aged 3–5 days were immobilized by cold anesthesia and transferred individually into 3.5 mm dishes. After a recovery period of 10–30 min, they were individually recorded. The environmental luminance was 1000–1300 Lux white light. The yellow light used to activate NpHR was supplied by 590 nm LEDs at an intensity of 0.06–0.1 mW/mm$^2$. An infrared source and a web camera with a visible light filter were used to record the movements of the flies. Each fly was recorded for 2 min. For each fly, the distance moved, the time spent moving and the speed during the second min were analyzed. Flies that did not move during the 2 min were excluded. The temperature was 24–26°C and the humidity was 40–60%. The locomotor assay in Drosophila are described in more detail at Bio-protocol (*Liu et al., 2017*).

## Quantitative Real Time PCR

The effectiveness of all innexin RNAi lines were verified with Quantitative Real Time PCR (qPCR). Total RNA extracted from approximately 100 fly heads with TRIzol (Invitrogen, Carsbad, California) was treated with RQ1 DNase (Promega, Fitchburg, Wisconsin) and reverse-transcribed using Prime-Script RT Master Mix (Takara, Japan). Relative quantification PCR was accomplished using a SYBR

Premix Ex TaqTM II kit (Takara, Japan) and an ABI PRISM 7300 real-time PCR Detection system (Applied Biosystems, Waltham, Massachussetts). Rp49 was used as an internal control and relative mRNA levels were calculated with the comparative CT method. Gene expression levels of all manipulated flies (elav-Gal4/UAS-inx RNAi) and control flies (UAS-inx RNAi/+) were normalized to elav-GAL4/+. Three separate samples were collected from each condition and measurements were conducted in triplicates.

## Electrophysiology

Whole-cell recordings were performed under visual control using a Zeiss Axioskop2 microscope or an Olympus BX61WI microscope with a 40× water-immersion objective. The recordings were performed at 20–25°C with a Multiclamp 700B amplifier (Molecular Devices, Sunnyvale, California). The data were low-pass filtered at 10 kHz and acquired at 20 kHz with a Digidata 1440A digitizer (Molecular Devices, Sunnyvale, California).

Fly brains were dissected out and immersed in extracellular saline solution in a small dish. The saline composition was as follows: 103 mM NaCl, 3 mM KCl, 5 mM N-Tris (hydroxymethyl) methyl-2-aminoethane-sulfonic acid, 10 mM trehalose, 8 mM glucose, 26 mM $NaHCO_3$, 1 mM $NaH_2PO_4$, 1.5 mM $CaCl_2$, and 4 mM $MgCl_2$, adjusted to 275 mOsm(*Wilson and Laurent, 2005*). The saline was bubbled with 95% O2/5% CO2 gas for a final pH of 7.3. The sheath covering the target neuron was removed with fine forceps. For neuron targeting, four GAL4 lines (OK107, MB247, c205, and c305a) were used to drive EGFP expression in KCs, and two split-GAL4 lines (MB002B and MB011B) were used to label $\beta'$2mp neurons. Patch-clamp electrodes (20–30 MΩ) were filled with the following solution: 140 mM potassium aspartate, 10 mM HEPES, 1 mM KCl, 4 mM MgATP, 0.5 mM $Na_3$GTP, 1 mM EGTA, and 100 μM Alexa Fluor 568 (Life Technologies, Carsbad, California) or 1% biocytin hydrazide, pH 7.3, adjusted to 265 mOsm. In the majority of the recorded neurons, a small constant hyperpolarizing current (-50–0pA) was injected to keep the membrane potential between −60 and −50 mV. In the drug delivery experiments, the final concentrations of TTX (Taizhou Kang Te, China) and 2-OCT were 1 μM and 1 mM, respectively. All reagents were purchased from Sigma-Aldrich, unless stated otherwise.

## Dye coupling

Low-molecular-weight dyes (biocytin and Alexa Fluor 568) were loaded into neurons through an electrode during whole-cell recording. Only one soma was successfully recorded per brain or hemisphere. For each brain, 1~4 attempts were made to get successful recording. Brains without successful recording were discarded after 4 attempts. No obvious dye in the target soma was confirmed by fluorescence after failed attempts. After loading for one to two hours, the dye labeled the soma and neurites of the recorded neuron and diffused to other neurons through open gap-junction channels if present. Because of the small size of KC somas, they were almost always removed with the electrode when the electrodes were drawn out after dye loading was complete. Thus the loaded soma was observed in only a few brains. Next, the brain was fixed in 4% PFA for 12–15 min at room temperature and then washed several times in PBS with 0.3% Triton X-100. The Alexa Fluor 568-labeled brains were mounted in Vectashield H-1000 (Vector Laboratories, Burlingame, California) for confocal imaging. The biocytin-labeled brains were incubated with Alexa Fluor 568-conjugated streptavidin (Life Technologies; 1:200) overnight at 4°C. The brains were washed several times and then mounted in Vectashield H-1000 for confocal imaging. A Nikon Eclipse FN1 confocal system with a 40× water-immersion objective and an Olympus FV10I system with a 60× oil-immersion objective were used to obtain the Z-stacks of the brains. The dye-filled neurites were identified based on their position and morphology. The examples that meet the following criteria were counted as dye-coupled examples: (1) More than one dye-filled fibers were observed in peduncle. (2) Only one fiber was observed in peduncle, but fibers in lobe region belonged to different subtypes. (3) Only one fiber was observed in the peduncle and fibers in the lobe region belonging to the same subtype, but were not branches of one neuron. Condition (3) only applied to $\alpha\beta$ core neurons, which generally have few branches. In blind experiments, the brain mounting, confocal imaging and identification of dye coupling were performed by an experimenter who was unaware of the genotypes of the brains.

## Statistical analysis

Data are presented as the mean ± SEM or by box plot. We applied the Wilcoxon signed rank test to compare the performance indices against chance level (*Brembs and Wiener, 2006*; *Liu et al., 1999*), and to evaluate differences between matched samples. Wilcoxon rank sum tests were used for comparisons between two groups. To compare three or more matched groups, we used Kruskal-Wallis test, and the significance of differences between each pair of groups were corrected by Bonferroni method. Statistical analysis was performed using MATLAB. Asterisks indicate the level of statistical significance ($*p < 0.05$; $**p < 0.01$; $***p < 0.001$).

## Acknowledgements

We sincerely thank Yoshinori Aso for assistance in identification of neurites, Mu-ming Poo for constructive comments, Mingkui Zhang and Jingwu Hou for assistance with experimental setup, and Liping Wang for assistance in optogenetics. This work was supported by the "Strategic Priority Research Program" of the CAS (XDB02040004), by grants from the 973 Program (2011CBA00400), as well as by the National Science Foundation of China (91232000, 91132709, 31130027, and 31070956).

## Additional information

### Funding

 No external funding was received for this work.

### Author contributions

QL, Designed this project, Established the optogenetic-flight simulator setup and conducted the behavioral experiments, Performed the dye coupling experiments, Analyzed the data, Wrote the manuscript; XY, Designed this project, Assisted with the behavior experiments, Performed the dye coupling experiments, Performed the electrophysiological experiments, Analyzed the data, Wrote the manuscript; JT, MW, Assisted with the behavior experiments, Analysis and interpretation of data; ZG, Performed the qPCR experiments, Analysis and interpretation of data; YL, AG, Designed this project, Wrote the manuscript

### Author ORCIDs

Xing Yang, http://orcid.org/0000-0001-6710-0012
Aike Guo, http://orcid.org/0000-0002-6515-7944

## Additional files

### Supplementary files

• Supplementary file 1. The specificity and target sequences of RNAi for *innexin* 1~8. THU, Tsinghua University.

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
