## [Decision Letter]

Thank you for submitting your work entitled "Gap Junction Networks in Mushroom Bodies Participate in Visual Learning and Memory in *Drosophila*" for consideration by *eLife*. Your article has been reviewed by two peer reviewers, including Benjamin de Bivort, and the evaluation has been overseen by K VijayRaghavan as the Senior Editor.

The reviewers have discussed the reviews with one another and the Reviewing Editor has drafted this decision to help you prepare a revised submission.

Summary:

In this paper, Liu and colleagues examine an important and understudied question in neuroscience, what is the role of gap-junctions in behavior, specifically visual pattern learning in *Drosophila*. The authors first reduce expression of a series of innexins, the genes encoding gap junctions, in the mushroom bodies, integrating neuropils which the authors characterize as the cognitive centers of the fly brain. They find that knocking down a specific subset of innexins renders flies' learning scores statistically indistinguishable from zero (no learning). Interestingly, blocking synaptic release using the dominant negative dynamin allele *shibire* has no effect on learning, suggesting that MB function in this behavior is gap-junction mediated. In a careful sequence of experiments and controls they are able to show that this effect is specific to regional subsets of the mushroom body and that *innexin* function is required post-developmentally. Next, the authors report diffusion of small dye molecules from patched Kenyon cells (KCs = MB intrinsic neurons) to other KCs. Using optogenetic stimulation paired with whole-cell recording, they show that hyperpolarization can be transmitted between KCs and a specific class of MB output neurons (MBONs), which can be temporarily blocked by the application of 2-octanol, a gap-junction blocker. Lastly, they find corresponding dependence of visual learning on innexin expression in those MBONs.

In general, the paper presents compelling genetic evidence of the requirement of MB-specific gap junction expression for visual learning, and direct evidence of gap junctions by dye diffusion (though as the authors discuss, the observed rate of dye coupling may underestimate the rate of gap-junctioning between pairs of KCs). The paper is well written, with generally solid use of controls. The balance of evidence is convincing that gap junctions likely play an important role in visual learning in the MBs.

Essential revisions:

While this is a very nice study, we do have a major conceptual concern and other important ones too.

1) Previous literature, including from the senior author of this study, has argued that MBs are totally dispensable for flight simulator visual learning except in cases where context is altered or where more complex combinations of features (such as shape and color) are used. This previous literature made use of HU ablation of MB. In other words, flies that have no MB structures at all are able to perform normally in this visual learning paradigm. Defects in visual learning in MB-less flies are only revealed with more complex tasks that were not employed here. This literature includes several classic studies from the Heisenberg group (e.g. Wolf et al. 1998 and Liu et al. 1999), as well as a more recent study by Guo and colleagues (Peng et al., 2007). So if in fact MB are not required for this task, it would obviously undermine the very foundation upon which they test the requirement for gap junctions within MB for this task. The authors correctly point out that a recent study by Vogt et al. points to a role for MB (using *shibire* to block output) in a visual learning task, and they correctly point to the literature that MB are required for visual learning that involves changes in context. But still, in this flight simulator visual learning assay, the literature is very consistent in the finding that MB ablated animals have no defect. The authors suggest that this discrepancy could be caused by developmental compensation with HU ablation. And we suppose that methodological differences in the flight simulator assay employed here could differ from those circa late 1990s. But this is the crux of the current study, so we were really surprised that they didn’t just do a simple HU ablation and test performance in their current assay. If performance is normal, it would be tough to read past the first figure without further explanation right? We see two types of explanation to reconcile the fact that HU ablation in previous publications has no impact on flight simulator visual learning, whereas acute manipulations of gap junctions does. One possibility is that when MB are ablated developmentally, there is some compensation mechanism in which the brain arrives at a solution that is independent of MB. This is certainly possible. But the other possibility is that the exact details of the current version of the flight simulator behavioral paradigm are sufficiently different from those historically reported. In this case, the current version of the flight simulator, in these researcher's hands, engages the MB in a way that previous versions of the assay did not. These are two really different interpretations. And it would be trivial for the authors to address this: just do an HU ablation of MB and test learning in their flight simulator. This lab has done HU ablation in the past. So this requires no new expertise and is such an obvious experiment that we would guess they have already thought of it. We agree that a positive result here would be an important addition to this study. If the authors find again that developmental ablation of the MB does not impair this behavior, then an in-depth discussion of the plausibility of developmental compensation is needed.

2) Our other major criticism concerns the presentation of statistical findings. Specifically, the authors conflate two statistical observations: that 1) an experimental group is not significantly different from 0 (while its control group is) and 2) that experimental group is therefore significantly different from the control. This faulty inference is made in several places. Relatedly, the author's description of the steps they took to correct for multiple comparisons is insufficient (but it is important to get this right, especially in Figure 1).

3) Next, we found the Introduction to be too short to provide proper background for the findings. It consists of a paragraph about the general function of gap junctions and a paragraph about the role of the mushroom bodies in learning and memory in flies. A major area for which there are previous findings is the role of gap-junctions in other *Drosophila* circuits, such as the optic lamina, the Horizontal system cells, and the giant fiber system.

4) Lastly, we would recommend that the authors edit their manuscript to make its interpretation easier for non-*Drosophila* MB experts. Specifically, Gal4 reagents are introduced without any description of their expression pattern. This makes interpreting the findings much harder.

5) In addition, we have few data presentation issues:

A) Figure 3—figure supplement 2: I cannot see the green fills in this figure. Can the images be improved?

B) It is stated that independent RNAi lines were used from different sources, but we could not find any information on the sequences used in the constructs. Were these derived from and targeting different regions of the mRNA? if not, they are not independent in a meaningful way.

3) We did not see any detail of the methods for RT-PCR to confirm that the RNAi lines work (Figure 2—figure supplement 1). Was this real time – PCR? End point PCR? PCR details? Are the effects reported as delta-delta CT values? Etc., and no stats were shown for this panel.

---

## [Author Response]

*Essential revisions:*

*While this is a very nice study, we do have a major conceptual concern and other important ones too.*

*1) Previous literature, including from the senior author of this study, has argued that MBs are totally dispensable for flight simulator visual learning except in cases where context is altered or where more complex combinations of features (such as shape and color) are used. This previous literature made use of HU ablation of MB. In other words, flies that have no MB structures at all are able to perform normally in this visual learning paradigm. Defects in visual learning in MB-less flies are only revealed with more complex tasks that were not employed here. This literature includes several classic studies from the Heisenberg group (e.g. Wolf et al. 1998 and Liu et al. 1999), as well as a more recent study by Guo and colleagues (Peng et al., 2007). So if in fact MB are not required for this task, it would obviously undermine the very foundation upon which they test the requirement for gap junctions within MB for this task. The authors correctly point out that a recent study by Vogt et al. points to a role for MB (using shibire to block output) in a visual learning task, and they correctly point to the literature that MB are required for visual learning that involves changes in context. But still, in this flight simulator visual learning assay, the literature is very consistent in the finding that MB ablated animals have no defect. The authors suggest that this discrepancy could be caused by developmental compensation with HU ablation. And we suppose that methodological differences in the flight simulator assay employed here could differ from those circa late 1990s. But this is the crux of the current study, so we were really surprised that they didn’t just do a simple HU ablation and test performance in their current assay. If performance is normal, it would be tough to read past the first figure without further explanation right? We see two types of explanation to reconcile the fact that HU ablation in previous publications has no impact on flight simulator visual learning, whereas acute manipulations of gap junctions does. One possibility is that when MB are ablated developmentally, there is some compensation mechanism in which the brain arrives at a solution that is independent of MB. This is certainly possible. But the other possibility is that the exact details of the current version of the flight simulator behavioral paradigm are sufficiently different from those historically reported. In this case, the current version of the flight simulator, in these researcher's hands, engages the MB in a way that previous versions of the assay did not. These are two really different interpretations. And it would be trivial for the authors to address this: just do an HU ablation of MB and test learning in their flight simulator. This lab has done HU ablation in the past. So this requires no new expertise and is such an obvious experiment that we would guess they have already thought of it. We agree that a positive result here would be an important addition to this study. If the authors find again that developmental ablation of the MB does not impair this behavior, then an in-depth discussion of the plausibility of developmental compensation is needed.*

We thank the reviewers for the comments and suggestions. We have performed the HU ablation experiments and found that ablation of the MBs at early developmental stage does not impair fly’s visual learning and memory, which is in agreement with previous reports (Liu et al., 1999; Peng et al., 2007; Wolf et al., 1998). We have strengthened the discussion of developmental compensation in our manuscript.

2) Our other major criticism concerns the presentation of statistical findings. Specifically, the authors conflate two statistical observations: that 1) an experimental group is not significantly different from 0 (while its control group is) and 2) that experimental group is therefore significantly different from the control. This faulty inference is made in several places. Relatedly, the author's description of the steps they took to correct for multiple comparisons is insufficient (but it is important to get this right, especially in Figure 1).

Thanks for pointing out our mistakes. We have rephrased our expression. When the performance index of the test group was significantly different from the control, we describe the visual learning of these flies as they were impaired. When the performance index of the test group was not significantly different from the control, and significantly different from zero, we describe the visual learning of these flies as they were intact. In Figure 2, the indices of 17d > *inx4* RNAi and 17d > *inx5* RNAi flies was nether significantly different from the control nor from zero. We changed the description of “knockdown of *inx4, inx5* and *inx6* in 17d-GAL4-labeled neurons resulted in impairment of fly learning and memory” to “knockdown of *inx6* in 17d-GAL4-labeled neurons resulted in impairment of fly learning and memory. Flies with the knockdown of *inx4, inx5* or *inx6* were not able to achieve performance indices significantly different to zero, whereas the control groups did.” We have also added the description of the correction for multiple comparisons in the Materials and methods.

*3) Next, we found the Introduction to be too short to provide proper background for the findings. It consists of a paragraph about the general function of gap junctions and a paragraph about the role of the mushroom bodies in learning and memory in flies. A major area for which there are previous findings is the role of gap-junctions in other Drosophila circuits, such as the optic lamina, the Horizontal system cells, and the giant fiber system.*

Thanks for the suggestions. We have added the introduction of studies on gap junctions in *Drosophila* circuits, like the giant fiber, the optic lamina, olfactory glomeruli and the APL-DPM neural circuits.

*4) Lastly, we would recommend that the authors edit their manuscript to make its interpretation easier for non-Drosophila MB experts. Specifically, Gal4 reagents are introduced without any description of their expression pattern. This makes interpreting the findings much harder.*

Thanks for pointing out our negligence. We have added the description of expression pattern of each Gal4.

5) In addition, we have few data presentation issues:

A) Figure 3—figure supplement 2: I cannot see the green fills in this figure. Can the images be improved?

We have uploaded images with higher resolution and stacks of images as videos. We hope these could improve the presentation.

B) It is stated that independent RNAi lines were used from different sources, but we could not find any information on the sequences used in the constructs. Were these derived from and targeting different regions of the mRNA? if not, they are not independent in a meaningful way.

Thanks for pointing out our negligence. We have added the description of the target sequences in [Supplementary-material SD1-data].

3) We did not see any detail of the methods for RT-PCR to confirm that the RNAi lines work (Figure 2—figure supplement 1). Was this real time – PCR? End point PCR? PCR details? Are the effects reported as delta-delta CT values? Etc., and no stats were shown for this panel.

Thanks for this comment. We have added the description of the methods for RT-PCR in the Materials and methods.